# Enhanced fibrinolytic enzyme production by *Oidiodendron maius* through green bioprocessing of agro-industrial residue

Hina Sajid[1], Muhammad Iqbal[2], Gull-e-Faran [1]*

1 Department of Biochemistry and Molecular Biology, Institute of Biochemistry, Biotechnology and Bioinformatics, The Islamia University of Bahawalpur, Bahawalpur, Pakistan, 2 Department of Environmental Sciences, Government College University Faisalabad, Faisalabad, Pakistan

* gullfaraniub@gmail.com

## Abstract

Thrombosis denotes the formation of blood clots within arteries and veins, representing a primary etiological factor in cardiovascular diseases often culminating in fatal outcomes. Prompt resolution of thrombotic disorders is achieved through expedited fibrinolysis facilitated by the administration of fibrinolytic enzymes, which constitute the optimal therapeutic approach. This research aimed to enhance the production of fibrinolytic enzymes through the cultivation of indigenously isolated strains of *Oidiodendron maius* using physical and chemical mutagenesis techniques. Fibrinolytic enzyme activity from mutant strains was validated by enzyme assays followed by purification using ammonium sulfate precipitation, desalting, ion exchange chromatography, gel filtration chromatography, and SDS-PAGE. Various kinetic and thermodynamic parameters were systematically optimized to maximize enzyme activity. In *O. maius*, the ethidium bromide mutant strain showed better results as compared to the other mutants with specific activity of 1642.24 U/mg and 0.5 mg/mL protein content compared to the wild-type strain which 90.20 U/mg specific activity and 3.7 mg/mL protein content. The optimum temperature and pH were 35°C and 7.5, respectively. The findings indicated that treating *O. maius* with ethidium bromide resulted in the generation of better mutants with enhanced enzyme activities compared to wild-type and other mutant strains. With optimization of multiple parameters, these strains demonstrate significant potential for enhanced fibrinolytic enzyme production by the usage of wheat bran as substrate.

## Introduction

The insoluble fibrin within blood vessels and abnormal formation of blood clots lead to thrombosis, known as thrombotic diseases (TDs), including myocardial infraction, stroke and venous thromboembolism [1]. Approximately 31% of global mortality is attributed to TDs, with the number gradually increasing in developing countries [2].

**Data availability statement:** All relevant data are within the manuscript.

**Funding:** The author(s) received no specific funding for this work.

**Competing interests:** The authors have declared that no competing interests exist.

Due to excess thrombin production under pathological conditions and uncontrolled hydrolysis of fibrinogen (FIB) to fibrin, insoluble fibrin fibers can precipitate in blood vessels, posing a significant challenge for clinical treatment [3]. Hence, effective thrombolytic agents are required for the treatment of TD patients.

Various proteases, including fibrinolytic enzymes, are responsible for degrading fibrin in thrombi. Fibrinolytic enzymes are categorized into two main types based on their mechanism of action. The first type includes plasminogen activators, such as urokinase, which hydrolyze plasminogen into plasmin, and tissue-type plasminogen activator (t-PA). The second type includes fibrin(ogen)olytic enzymes, such as lumbrokinase and nattokinase [4,5]. These fibrinolytic enzymes have been clinically reported as effective thrombolytic agents. However, they present various side effects, including excessive bleeding, low fibrin specificity, short half-life, and other clinical limitations [6]. Therefore, there is critical need for research focused on identifying efficient thrombolytic agents from natural sources, including agro-industrial residues.

*O. maius* is classified as an ericoid mycorrhiza fungus, known to form a symbiotic association with plant roots. This relationship facilitates nutrients exchange between the plant and the fungi. The fungus is reported to synthesize fibrinolytic enzymes capable of hydrolyzing complex proteins into simpler amino acids. It has also been found in peat soil and compost in several regions around the world [7].

The aim of the present study was to achieve hyperproduction of fibrinolytic enzymes from *O. maius*. Mutations were induced in wild-type *O.* maius using chemical and physical agents to promote fibrinolytic enzyme hyper-production. Characterization and purification of these enzymes were followed by optimization of the production medium to enhance yield.

## Materials and methods

All the chemicals used in the materials and methods section were of sigma grade.

### Isolation of *O. maius*

A number of compost and soil samples from different field areas of Bahawalpur City, Pakistan were collected. The compost and soil samples were air-dried and crushed into fine powder. The standard incubation time and temperature were applied to the collected samples to promote fungal grow [8]. The serial dilution method was used to prepare a series dilutions of soil and compost samples. The morphological characterization of *O. maius* was conducted following the protocol of Rice and Currah (2005) [8]. Colonies were measured after two weeks. Microscopic examination was performed for detailed phenotypic examination. Cotton blue staining with Lacto-Phenol and colony characteristics were used to evaluate fungal growth. The confirmation of observed results was done by comparing with the results reported by Rice and Currah, 2005.

*O. maius* was tested for hemoglobin hydrolysis on PDA media supplemented with 0.2% bovine hemoglobin. After autoclaving and solidification, the media was inoculated with *O. maius* and incubated in the dark for 12–13 days. Hemoglobin hydrolysis was detected using two methods. Zones of clearance were observed after washing

with 2M HCl and 12% HgCl$_2$. Clear zones were visible after washing with 30% methanol, 10% acetic acid, and 0.1% amido black. Clear zones indicated proteolytic activity.

Modified Melin Norkrans (MMN) medium was prepared with the following composition: CaCl$_2$ (1 g), KH$_2$PO$_4$ (10 g), malt extract (3.5 g), NaCl (0.5 g), D-glucose anhydrous (1 g), agar (12 g), MgSO$_4$·7H$_2$O (3.1 g) and 1 liter of double-distilled water. The fungal isolates were cultivated MMN medium for five weeks, supplemented with citrus pectin (5g/L, Sigma grade), to assess pectinase activity. After incubation, the plates were flooded with 1% aqueous solution of hexadecylmethylammonium bromide to visualize clear zones.

Cellulose azure was used to evaluate cellulose activity, again following Rice and Currah (2005). In Pyrex tubes, 20 mL of MMN solution was added and autoclaved, followed by solidification at room temperature. A 2% (w/v) cellulose azure solution (Sigma-Aldrich) was poured over solidified MMN agar media plates containing fungal mycelia. After 3–5 weeks, results were recorded. Lipase activity was assessed using MMN medium containing Tween-20 (10 mL/L) and CaCl$_2$ (0.1 g/L). For detecting lipase synthesis by mycelia, fungal isolated were incubated on media plates for 4–6 weeks. Macroscopic results were recorded and scored as positive after the incubation period.

MMN medium was also prepared in petri plates using 65 g/L gelatin, replacing agar. After autoclaving, gelatin was dissolved in 900 mL distilled water. The plates were inoculated with fungal isolates and incubated for 3–5 weeks. Positive results were recorded by observing gelatin degradation in the petri dishes. Sabouraud dextrose agar (SDA) was selected as the subculture medium for *O. maius*. All the tests were performed in triplicates along with appropriate controls.

## Strain improvement by mutagenesis

Gamma and Ultra-Violet (UV) irradiation were employed to induce mutations in *O. maius* for the hyperproduction of fibrinolytic enzymes. Gamma irradiation has been reported to be more mutagenic than chemical mutagens [9].

## Preparation of conidial suspension

Conidia suspensions were prepared following the method reported by Hameed *et al.*, [10]. Petri plates containing Potato Dextrose Agar (PDA) medium were flooded with double-distilled water. To remove mycelial fragments, sterile cotton was used for filtration the resulting suspensions. Serial dilution was performed in the test tubes to obtain a concentration range of 100–200 CFU/mL. Flasks containing 25 mL double-distilled water were prepared in duplicates for use as parental control and for mutagenesis testing.

## Physical mutagenesis

Gamma irradiation (Co$^{60}$) was employed for the physical mutagenesis. After a 7-day incubation on PDA media at 37°C, 0.05% Tween-80 was used to harvest fungal spores. A spore suspension of $1 \times 10^6$ spores/mL in McCartney vials) was prepared, and 10 mL of each suspension was subjected to Co$^{60}$ gamma irradiation. Seven gamma doses (20, 40, 60, 80, 100, 120 and 140 KRad) were tested [11]. A dose of 120 KRad was identified as the optimal level for *O. maius*, where 3-log kill curves were observed. UV radiation was also used to induce physical mutations in *O. maius*, with slight procedural modifications. Spore suspensions were poured into petri dishes, which were exposed to UV light at a wavelength of 254 nm for one hour, with 5-minute exposure intervals, maintaining a distance of 10 cm from the UV source.

## Chemical mutagenesis

Fungal spores were cultivated on PDA plates and incubated at 30 ± 1°C for 5–7 days until sporulation occurred. After fermentation, all mutagenized colonies were screened based on enzyme activity. A stock solution of 0.50 mg/mL ethidium bromide was prepared in conical flask, and 1 mL of it was added to 9 mL of fungal spore suspension. The mixture was centrifuged at 10,000 rpm for 15 minutes after incubation at varying intervals (15–210 minutes). A 150-minutes incubation period was selected as optimal for *O. maius.* Sodium nitrate solution (0.1M) in 50mM sodium acetate buffer (pH 5.0)

                                                                 

was prepared and 1mL of the reagent was added into the spore suspension (9mL) of *O. maius* for 15, 30, 45, 60, 75 and 90 minutes at 28°C. Then the reaction tubes were centrifuged at 10,000 rpm for 5 min. Pellets were washed with distilled water followed by inoculation over PDA plates. The 60 minutes dose was selected for *O. maius* as best killing rate giving dose.

## Mutated strain selection

Suspensions of mutated cells from four mutagen types were prepared, and 100-fold serial dilutions were made. A non-irradiated inoculum served as a control. Colonies were grown on PDA media plates with a colony restrictor, triton X-100 (1%), for 3–5 days at 30°C, covered with Aluminum foil. Enzyme activities of selected colonies were evaluated as described in [12]. Selected mutated strains were used for inoculum preparation, using 50 ml phosphate buffer with pH 7.0 (K$_2$HPO$_4$; 16.2g, KH$_2$PO$_4$; 9.79g, Distilled water; 1000g) [13]. To prepare the production medium, the following mixture in 1 liter of distilled water: Fibrin (2 g/l), NH$_4$NO$_3$ (0.05 g/l), KH$_2$PO$_4$ (1 g/l), (NH$_2$)$_4$SO$_4$ (0.5 g/l). All the chemicals used were of Sigma grade. Agro-industrial wheat bran was added at 5% (w/v) as a low cost substrate to enhance fibrinolytic enzyme yield via liquid-state fermentation. A 5% inoculum was aseptically transferred into individual production flasks and incubated 37°C on a shaker at 120 rpm for 12–72 hours.

## Parameter optimization

Parameter optimization was performed by following the protocol of Dr. Shilpa H.K. [14]. Variables such as: substrate concentrations (1–10%), temperature (25–60°C), fermentation time (12–72 hr.), pH (4.0–9.0), inoculum size (1.5–5.0 mL), were tested to determine conditions for maximum enzyme yield [15]. Supernatants were tested for proteolytic activity using a fibrin plate assay with slight modifications, as described by Vijayaraghavan and Samuel [16].

## *In vitro* procedure for clot lysis assay of fibrinolytic enzymes produced indigenously

Fibrin plates were prepared containing 0.75% (w/v) fibrinogen, 100 NIH U/mL thrombin and 1.3% (w/v) agarose, adjusted to pH 7.4 and incubated at 37°C for one hour to allow the formation of the fibrin clot layer. A sterile cork borer was used to puncture 5mm diameter wells, which were then filled with crude enzymes extracts. To obtain the enzyme crude extract, the fermentation broth was centrifuged at 12,000 rpm for 15 minutes to separate the supernatant. The fibrin clot assay was carried out by incubating the prepared petri plates at 37°C for 16–18 hours in triplicates with a control. Fibrin degradation around the wells, resulting in clear zones confirmed the thrombolytic potential of the enzyme isolated from the mutated strain [16–18].

## Protein contents estimation

Bradford assay was employed to estimate protein content following the method described by Bradford [19].

## Enzyme purification

The fibrinolytic enzymes were purified from crude enzyme by precipitation technique using ammonium sulfate salt (40% to 60%) followed by dialysis. To remove ammonium sulfate and other impurities from the partially purified enzyme, dialysis was performed for 4 hours using a dialysis membrane against double distilled water. After these two purification steps, ion exchange chromatography, equilibrated with 20 mM Tris HCl buffer (pH 7.4), was performed and then crude enzyme was subjected to the sephadex G-50 gel filtration chromatography for final step purification.

## Sodium Dodecyl Sulfate Polyacrylamide Gel Electrophoresis (SDS-PAGE)

The molecular weight of the purified fibrinolytic enzymes was determined using SDS-PAGE and bands were excised from the gel as per the procedure outlined in [20]. The markers used in SDS-PAGE were of low molecular weight (Sigma).

## Fibrinolytic Activity of Purified Enzymes

The punch-hole method was used to assess the fibrinolytic activity [21]. Fibrin plates were prepared by adding 0.75 g of fibrin and 0.5 g of agar in distilled water, followed by heating at 80°C to dissolve the agar. After solidification at room temperature, 5 mm diameter holes were punched into the plates, and 50 µl of purified enzyme was added into each well. The plates were incubated at 37°C for 18 hours, and the transparent zones around the wells measured to assess enzymatic activity.

## CaCl$_2$-induced clotting time assay

A mixture of containing 0.8 mL of extracted enzyme, 1 mL of human blood plasma, and 0.2 mL of CaCl$_2$ solution was prepared. EDTA was used as a standard anticoagulant. Control samples without enzyme or EDTA were also prepared. The ingredients were gently mixed by inverting the test tubes to moisten the inner surface. Clotting time was immediately recorded for each test tube.

## Thermodynamic and kinetic parameters

The thermodynamic and kinetic optimization of the purified fibrinolytic enzyme was performed following the standard protocols reported in [22,23] to evaluate its catalytic efficiency under varying conditions.

## Effect of temperature on enzyme activity

The proteolytic activity of enzymes from both parental and mutated strains of *O. maius* was assessed at varying temperatures (20–80°C) at an optimized pH of 7.0. Enzyme assay method was used to monitor enzyme activity [15].

## Effect of pH on enzyme activity

Enzymes derived from mutated and parental *O. maius* strains were tested at different pH values (4.0–8.0) to determine the optimum pH. A 0.5 mL substrate of 1% casein was mixed with 0.5 mL of purified enzyme, incubated at 37°C for 30 minutes, and enzyme activity was evaluated.

## Effect of metal ions on fibrinolytic activity

The effects of metal ions (CuSO$_4$, FeCl$_2$, ZnCl$_2$, CaCl$_2$ and MgCl$_2$) were analyzed. Purified enzymes from parental and mutated strains were pre-incubated in the presence or absence of Fe$^{2+}$, Zn$^{2+}$ and Mg$^{2+}$ ions in 10 mM citrate-NaOH buffer (pH 6.0) at 37 °C for one hour. Enzyme activity was then measured [22].

## Effect of inhibitors

Various enzyme inhibitors were tested, including EDTA, PMSF (1–10 µM), 1, 10-phenanthroline, p-chloromercuribenzoate, hydrogen peroxide (1%), SDS (1%), and NaCl brine solution. The fibrinolytic activity of wild-type and UV-mutant *O. maius* enzymes was assessed using the method reported by Shilpa H.K. [14].

## Activation Energy ($E_a$)

Activation energy of the enzymes was determined over a temperatures range of 20–70°C by following the Arrhenius protocol [24]. Both mutated and parental enzyme samples were analyzed.

## Kinetic studies

To calculate the Michaelis constant (Km), fibrinolytic enzyme activity was tested at various concentrations of fibrinogen. A Lineweaver-Burk plot was constructed by plotting 1/V against 1/S, and Km was derived from the resulting linear regression [12,14].

 

## Statistical analysis

Data was statistically analyzed using a two-factor randomized design with ANOVA to assess the standard errors and means. Graphs were plotted using Microsoft Excel to visualize the results.

## Results

### Isolation of *O. maius* isolates for fibrinolytic enzyme production

Identification of the isolates was carried out using standard biochemical and morphological methods. Key morphological traits such as hyphal structure, growth pattern, colony color, surface texture, aerial mycelium, spore production mechanism, colony margin, and conidial characteristic were examined. Microscopic and biochemical characteristics were recorded and compared to a known reference culture, confirming the identity of selected strain as *O. maius* (Table 1 and 2). Fig 1 depicts the *O. maius* colony on SDA media.

### Colony growth restriction and mutant screening

To identify high-yielding fibrinolytic enzyme-producing strains, several screening assays were performed, including colony growth restriction, fibrin plate assay, mutant strain identification, and enzyme diffusion zone test. The outcomes of these tests were recorded and analyzed. Based on the zone of clearance in fibrin plates and enzyme diffusion assay, several strains were shortlisted as potential candidates for enhanced fibrinolytic activity. BBTI-UV-50, BBTI-G-120, BBTI-EB-150, and BBTI-NA-60 mutant strains of *O. maius* were screened and selected for their substantial enzyme production potential. These strains demonstrated promising results in synthesizing levels of fibrinolytic enzymes (Table 3).

### Production of enzymes

**Fibrinolytic enzyme production through liquid state fermentation.** Various fermentation strategies were employed to reduce enzyme production costs and improve yield. These included the selection of an ideal microbial strain, use of economical and reliable fermentation media, choice of suitable low-cost substrate, and the optimization of key

**Table 1. Microscopic studies of *Oidiodendron maius* from different sources.**

| Character | *BBTI-1 | BBTI-2 | BBTI-3 | BBTI-4 | BBTI-5 | Reference organism |
|---|---|---|---|---|---|---|
| Conidiomata | Absent | Present | Absent | Absent | Absent | Absent |
| Conidial color | Hyaline | Lightly pigmented | Hyaline | Hyaline | Pigmented | Hyaline |
| Conidiophore branching | Unbranched | Branched | Unbranched | Unbranched | Branched | Unbranched |
| Colony surface color | Pale brown | Yellow | Creamy yellow | Yellow | Creamy yellow | Yellow |
| Conidiophore texture | Smooth | Minutely asperulate | Highly asperulate | Smooth | Highly asperulate | Smooth or minutely asperulate |
| Conidial shape | Subglobose | Elongate | Elongate | subglobose | Elongate | Subglobose, ellipsoidal or elongate |

*Source of morphology description of reference organism (Rice ad Currah, 2005).

*Bahawalpur Bio-Thrombolytic BBTI isolate 1, 2, 3, 4, 5.

**Table 2. Biochemical tests for different strains of *Oidiodendron maius*.**

| Sample Isolates | Source | Gelatinase activity | Cellulose Azure | Pectinase activity | Starch hydrolysis | Lipase synthesis | Lignin Degradation |
|---|---|---|---|---|---|---|---|
| BBTI-1 | Compost | + ve | +ve | +ve | +ve | +ve | -ve |
| BBTI-2 | Compost | -ve | +ve | -ve | -ve | +ve | -ve |
| BBTI-3 | Soil | +ve | +ve | +ve | +ve | +ve | -ve |
| BBTI-4 | Soil | +ve | +ve | +ve | +ve | +ve | -ve |
| BBTI-5 | Compost | +ve | +ve | -ve | +ve | -ve | -ve |

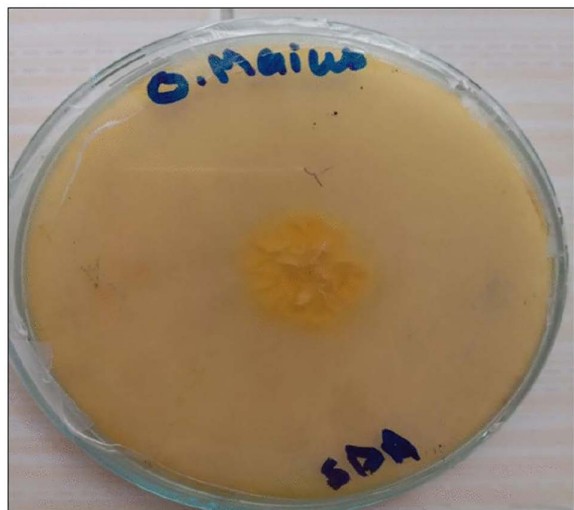

**Fig 1.** *Oidiodendron maius* **colony on SDA media.**

**Table 3.** Selected mutants of *Oidiodendron maius* after screening and selection.

| Strains | Fibrinolytic Enzyme Activity (U/mL) |
|---|---|
| *Oidiodendron maius* BBTI-UV-50 | 216.42 ± 1.84 |
| *Oidiodendron maius* BBTI-G-120 | 231.20 ± 3.76 |
| *Oidiodendron maius* BBTI-EB-150 | 242.13 ± 2.19 |
| *Oidiodendron maius* BBTI-NA-60 | 207.13 ± 1.34 |
| *Oidiodendron maius* (Wild) | 167.14 ± 1.08 |

fermentation period. The results demonstrated that fibrinolytic enzyme production through liquid state fermentation is highly dependent on the fine-tuning of these parameters.

**Parameter optimization for enzyme hyper-production.** **Optimization of pH** It was observed that both lower and higher pH levels adversely affected enzyme production compared to the optimum pH. The maximum enzyme production was recorded at the optimal pH, indicating the critical role of medium pH in influencing the synthesis of fibrinolytic enzymes. Similarly, the best enzyme yields for the wild-type and UV, Gamma, ethidium bromide, and nitrous acid mutated strains of *O. maius* were observed as 197.56, 231.98, 243.6, 278.05, and 224.07 U/mL at pH levels of 7.5, 7.0, 6.0, 8.0 and 7.0 respectively (Fig 2A). Compared to the wild-type fungal strain, all mutant strains demonstrated significantly improved enzyme production (Table 4).

## Optimization of temperature

Enzyme production for wild-type and mutant strains of *O. maius* was evaluated at temperature ranges of 25–60°C. For the UV, ethidium bromide, gamma, and nitrous acid mutants, maximum yields of 261.11, 336.91, 241.23, and 205.07 U/mL were observed at 35°C, 37°C, 37°C, and 40°C respectively. In contrast, the wild-type strain showed an enzyme yield of 194.32 U/mL at 37°C. These findings clearly indicate that the enzyme yield of mutant strains increased significantly at the optimum temperature of 37°C compared to the wild-type strain. Overall enzyme yield was notably affected by incubation temperature (Fig 2B). Statistical results also showed better enzyme production from the mutant strains after temperature optimization (Table 5).

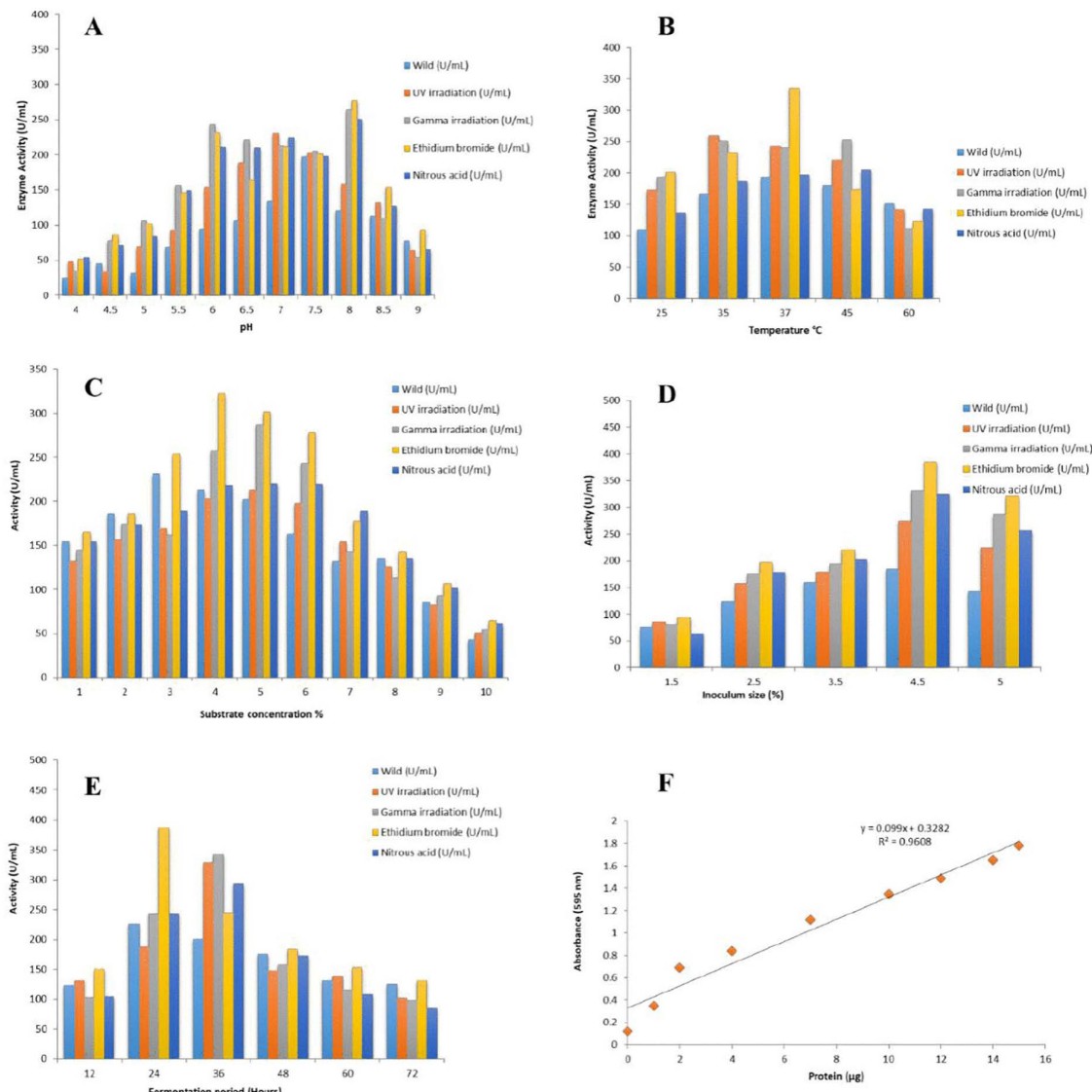

**Fig 2. Comparative enzyme production from wild-type and mutated *Oidiodendron maius* strains across A) varying pH values B) varying temperatures C) different substrate concentrations D) different inoculum sizes E) different fermentation periods F) Bradford standard curve for *Oidiodendron maius*.**

**Table 4. Analysis of variance table for optimization of pH for fibrinolytic enzyme production from wild and mutated strains of *Oidiodendron maius*.**

| Source of Variation | Sum of squares | Degrees of freedom | Mean squares | F-value | P-Value | F Crit |
|---|---|---|---|---|---|---|
| pH | 205326 | 10 | 20532.6 | 25.72 | 2.32267E-14 | 2.07 |
| Treatment | 32440.98 | 4 | 8110.24 | 10.16 | 0.0000089 | 2.60 |
| Error | 31921.81 | 40 | 798.04 | -- | -- | -- |
| Total | 269688.8 | 54 | -- | -- | -- | -- |

** =Highly significant (P<0.01).

## Optimization of substrate concentration

For the production of fibrinolytic enzymes, wheat bran was utilized as the substrate to grow *O. maius* wild-type and mutant strains. Among different substrate concentrations tested, the best enzyme yields were observed as U/mL (wild-type strain at 3%), 213.43 U/mL (UV mutant at 5%), 287.07 U/mL (gamma mutant at 5%), 322.35 U/mL (ethidium bromide mutant at 4%), and 220.03 U/mL (nitrous acid mutant at 5%) (Fig 2C). These findings indicate a variation in optimal substrate concentration for maximum fibrinolytic enzyme production depending on the type of strain. The ANOVA results (Table 6) clearly indicate that the maximum yields of fibrinolytic enzymes were achieved with the addition of 4% and 5% of substrate concentration.

## Optimization of inoculums size

In this study, the effect of different inoculum sizes (1.5, 2.5, 3.5, 4.5, and 5.0 mL) was evaluated in the growth media to determine the optimum level for fibrinolytic enzyme production. Enzyme synthesis by wild-type and UV, Sodium azide, Gamma, and ethidium bromide-treated *O. maius* mutants were observed to increase with the addition of 4.5% inoculum size (Fig 2D). Statistical results presented in Table 7 also support these findings.

## Optimization of fermentation period

Fibrinolytic enzyme production was carried out using wild-type, UV-, gamma-, ethidium bromide-, and nitrous acid-exposed strains of *O. maius* in liquid state production media for 12, 24, 36, 48, 60, and 72 hours. For the wild-type, UV, gamma, ethidium bromide, and nitrous acid-exposed strains, the maximum enzyme yields were recorded as 226.10, 329.34, 342.52, 387.11 and 293.23 U/mL at 24, 36, 36, 24 and 36 hours, respectively (Fig 2E). The findings demonstrated the enzyme production with longer fermentation time up to the optimum point (Table 8).

**Table 5. Analysis of variance table for optimization of temperature for fibrinolytic enzyme production from wild and mutated strains of *Oidiodendron maius*.**

| Source of Variation | Sum of squares | Degrees of freedom | Mean squares | F-value | P-value | F crit |
|---|---|---|---|---|---|---|
| **Temperature** | 38341.36 | 4 | 9585.34 | 9.42 | 0.00041 | 3.01 |
| **Treatment** | 14739.36 | 4 | 3684.84 | 3.62 | 0.027 | 3.01 |
| **Error** | 16267.84 | 16 | 1016.74 | -- | -- | -- |
| **Total** | 69348.56 | 24 | -- | -- | -- | -- |

** =Highly significant (P<0.01).

**Table 6. Analysis of variance for optimization of substrate concentration (%) for fibrinolytic enzyme production by wild-type and mutated *Oidiodendron maius* strains.**

| Source of Variation | Sum of squares | Degrees of freedom | Mean squares | F-value | P-value | F crit |
|---|---|---|---|---|---|---|
| **Substrate** | 177339.5 | 9 | 19704.4 | 33.73 | 6.32219E-15 | 2.15 |
| **Treatment** | 15810.15 | 4 | 3952.5 | 6.76 | 0.000361 | 2.63 |
| **Error** | 21030.1 | 36 | -- | -- | -- | -- |
| **Total** | 214179.8 | 49 | -- | -- | -- | -- |

** =Highly significant (P<0.01).

   

**Table 7. Analysis of variance for optimization of inoculum size for fibrinolytic enzyme production by wild-type and mutated *Oidiodendron maius* strains.**

| Source of Variation | Sum of squares | Degrees of freedom | Mean squares | F value | P-value | F crit |
|---|---|---|---|---|---|---|
| Inoculum size | 139068.3 | 4 | 34767.08 | 35.28 | 9.4524E-08 | 3.01 |
| Treatment | 31069.8 | 4 | 7767.6 | 7.88 | 0.00103 | 3.01 |
| Error | 15764.3 | 16 | 985.3 | -- | -- | -- |
| Total | 185902.49 | 24 | -- | -- | -- | -- |

** =Highly significant (P<0.01).

**Table 8. Analysis of variance for optimization of fermentation period for fibrinolytic enzyme production by wild-type and mutated *Oidiodendron maius* strains.**

| Source of Variation | Sum of squares | Degrees of freedom | Mean squares | F-value | P-value | F crit |
|---|---|---|---|---|---|---|
| Time | 137440.73 | 5 | 27488.15 | 15.86 | 0.000023 | 2.71 |
| Treatment | 7593.88 | 4 | 1898.47 | 1.09 | 0.0038 | 2.86 |
| Error | 34658.38 | 20 | 1732.9 | -- | -- | -- |
| Total | 179693.01 | 29 | -- | -- | -- | -- |

** =Highly significant (P<0.01).

## Sample harvesting

Total enzyme from liquid state production media of all strains flasks was harvested after the centrifugation at 12,000 rpm for 10 min at 4ºC. Filtration step was performed after centrifugation and supernatants were assayed for enzyme activity.

## Enzyme assay

**Protein estimation by Bradford assay.** Absorbance was measured at 595 nm using 96-well microtiter plate reader, and a standard curve was plotted between absorbance and protein concentration of each standard (Fig 2F). The protein contents in the crude extracts were calculated based on this standard curve. The results presented in the graph demonstrated a significant amount of protein present in the crude enzyme extracts from the fungal strains.

## Purification Profile of Enzymes from Wild vs Mutant fungal strains

**Ammonium sulfate precipitation for partial purification.** Fibrinolytic enzymes were synthesized under optimized conditions from physically and chemically derived mutants of *O. maius* and subsequently subjected to ammonium sulfate precipitation for partial purification. The results obtained for their crude enzyme activities, specific activities, protein contents, fold purification and percentage recoveries are depicted in Table 9. Similarly, the Figs 3A, B, C, D, and E show the results obtained for the enzyme activities, specific activities and protein contents from the supernatants and sediments acquired while performing 40% and 60% ammonium sulfate precipitation

## Desalting of fibrinolytic enzymes

The enzyme activities, specific activities, fold purification and recovery percentage of the respective strains were recorded after the desalting step and the deduced results are presented in Table 9 and Fig 3. After desalting, elevated levels of enzyme activity were observed in the wild-type and mutated strains (UV, gamma, ethidium bromide and nitrous acid treated) of *O. maius*.

**Table 9. Purification summary of fibrinolytic enzyme produced by wild-type and UV-mutated *Oidiodendron maius* Strains, Gamma mutated *Oidiodendron maius*, Ethidium bromide mutated *Oidiodendron maius* and nitrous acid mutated *Oidiodendron maius*.**

| Purification Stage | Activity (U/ml) | | Protein (mg/ml) | | Specific activity (U/mg) | | Fold Purification | | Recovery (%) | |
|---|---|---|---|---|---|---|---|---|---|---|
| | Wild | Mutant | Wild | Mutant | Wild | Mutant | Wild | Mutant | Wild | Mutant |
| **UV mutated *O. maius*** | | | | | | | | | | |
| **Crude** | 368 | 490 | 30 | 32.9 | 15.2 | 65.34 | 1 | 1 | 100 | 100 |
| **(NH₄)₂SO₄ Desalted** | 342 | 476 | 14.2 | 28 | 27.51 | 82 | 2.02 | 3.03 | 87.05 | 75 |
| **DEAE-Cellulose** | 309 | 347 | 5.2 | 12.34 | 44.72 | 382.45 | 3.57 | 11.25 | 77.82 | 75.43 |
| **Sephadex G-50** | 287 | 297 | 3.7 | 0.71 | 90.20 | 1379.82 | 7.02 | 69.24 | 73.15 | 77.73 |
| **Gamma mutated *O. maius*** | | | | | | | | | | |
| **Crude** | 368 | 495 | 30 | 18.52 | 15.2 | 32.2 | 1 | 1 | 100 | 100 |
| **(NH₄)₂SO₄ Desalted** | 342 | 371 | 14.2 | 10.2 | 27.51 | 73.72 | 2.02 | 3.14 | 87.05 | 93.34 |
| **DEAE-Cellulose** | 309 | 414 | 5.2 | 3.7 | 44.72 | 393.02 | 3.57 | 14.86 | 77.82 | 88.73 |
| **Sephadex G-50** | 287 | 372 | 3.7 | 1.01 | 90.20 | 1472.43 | 7.02 | 78.82 | 73.15 | 86.74 |
| **Ethidium bromide mutated *O. maius*** | | | | | | | | | | |
| **Crude** | 368 | 580 | 30 | 26.35 | 15.2 | 39.20 | 1 | 1 | 100 | 100 |
| **(NH₄)₂SO₄ Desalted** | 371 | 482 | 14.2 | 10.01 | 27.51 | 79.74 | 2.02 | 5.5 | 87.05 | 97.6 |
| **DEAE-Cellulose** | 309 | 342 | 5.2 | 1.80 | 44.72 | 436.37 | 3.57 | 16.02 | 77.82 | 90.87 |
| **Sephadex G-50** | 287 | 401 | 3.7 | 0.5 | 90.20 | 1642.24 | 7.02 | 78.91 | 73.15 | 71.09 |
| **Nitrous acid mutated *O. maius*** | | | | | | | | | | |
| **Crude** | 368 | 432 | 30 | 16.21 | 15.2 | 20.92 | 1 | 1 | 100 | 100 |
| **(NH₄)₂SO₄ Desalted** | 342 | 430 | 14.2 | 8.1 | 27.51 | 69.93 | 2.02 | 3.68 | 87.05 | 90.53 |
| **DEAE-Cellulose** | 309 | 392 | 5.2 | 1.92 | 44.72 | 332.55 | 3.57 | 14.37 | 77.82 | 82.42 |
| **Sephadex G-50** | 287 | 337 | 3.7 | 0.65 | 90.20 | 1482.42 | 7.02 | 78.36 | 73.15 | 68.34 |

### Ion exchange chromatography

The enzyme fractions obtained after ion exchange chromatography from the wild-type and mutant strains (UV, gamma, ethidium bromide, and nitrous acid) of *O. maius* were analyzed and the enzyme activities, specific activities, fold purification, protein contents and percentage recovery from the most active fraction for each strain were identified. Notably, the enzyme obtained from BBTI-EB-150 strain showed better performance, yielding 436 U/mg enzyme activity and 16.02-fold purification (Table 9).

### Purification by Gel filtration Chromatography

The most active fractions obtained from ion exchange chromatography were selected for further purification using gel filtration. The results calculated from the findings are concluded in Table 9.

### Sodium Dodecyl Sulfate Polyacrylamide Gel Electrophoresis (SDS-PAGE)

The purified enzyme samples were treated with mercaptoethanol and run on a 10% SDS-PAGE gel. The resulting gel displayed a prominent band with molecular weight of approximately 35 kDa (Fig 4).

### Assays of Enzymes for Clot lysis and Biomedical relevance

**Fibrinolytic activity of purified enzymes.** The clot hydrolysis assay was conducted to confirm the presence of fibrinolytic activity in the purified enzyme sample. This assay involved treating blood clots with enzymes obtained from

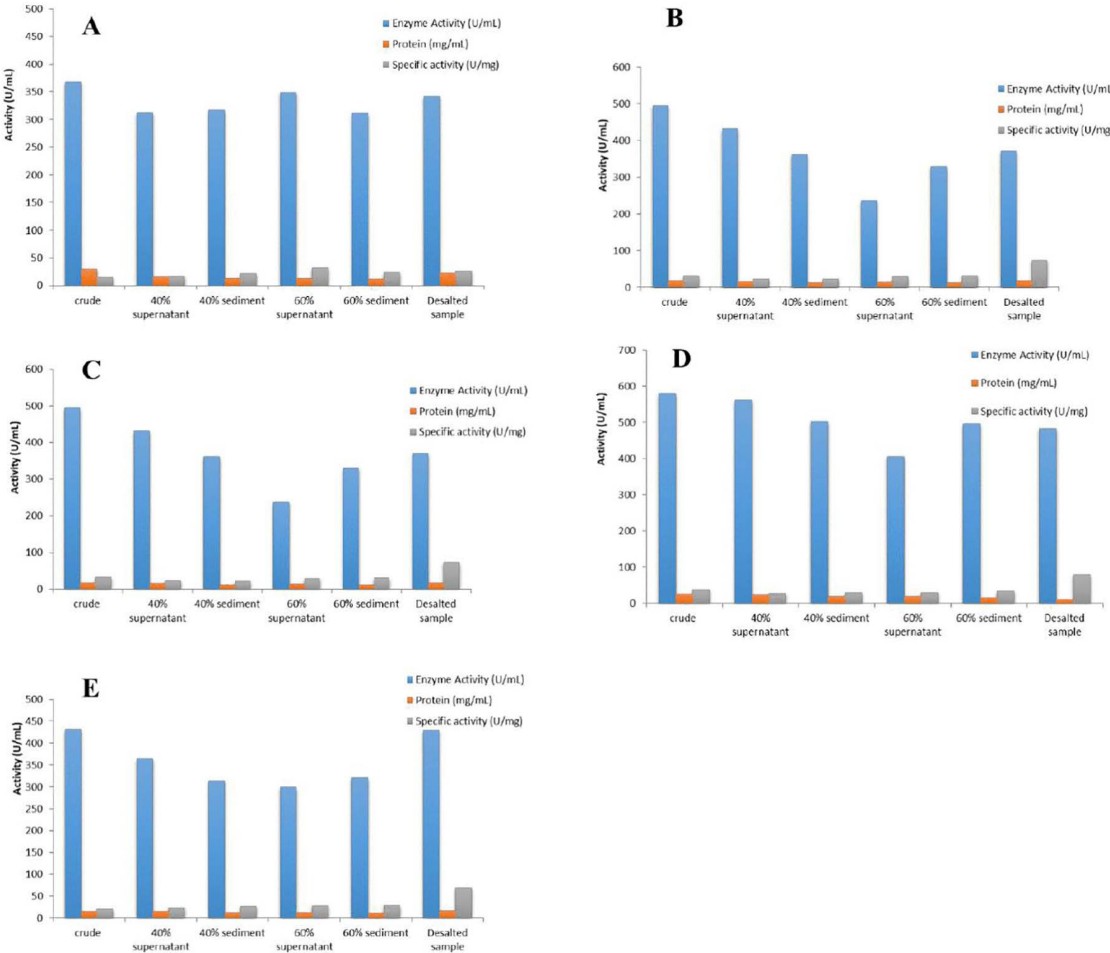

**Fig 3. Ammonium sulphate precipitation of fibrinolytic enzyme produced by A) wild-type *Oidiodendron maius* strain B) UV mutated *Oidiodendron maius* strain C) gamma mutated *Oidiodendron maius* strain D) Ethidium Bromide mutated *Oidiodendron maius* strain E) nitrous acid mutated *Oidiodendron maius* strain.**

mutant fungal strains. The lytic action of these fibrinolytic enzymes on the blood was analyzed (Fig 5). An increase in enzyme units corresponded with an increase in the percentage of clot lysis. Specifically, when 100 units of fibrinolytic enzyme from BBTI-EB-150 were applied, they exhibited 45% clot lysis. These results were also compared with standard enzyme controls.

### CaCl$_2$-induced Clotting Time Assay

The CaCl$_2$-induced clotting time assay was conducted to evaluate the anticoagulation activity of the fungal species. EDTA was used as the standard anticoagulant and exhibited higher anticoagulation activity compared to the fungal fibrinolytic enzymes, with clotting of plasma observed after 30 minutes. In contrast, the fibrinolytic enzymes derived from mutants strains of *O. maius* induced plasma clotting after 25 minutes. The assay was designed to determine the 50% clotting time and assess the effect on fibrin formation. The assay involved the addition of extracellular extracts to human plasma, followed by the addition of 0.16 M CaCl$_2$ to initiate clotting (Fig 6). The clotting time was recorded accordingly (Table 10).

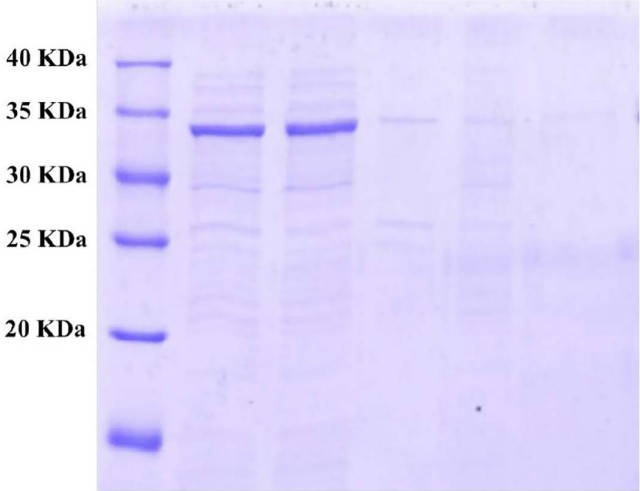

**Fig 4. SDS-PAGE evaluation of fibrinolytic enzymes produced by BBTI-EB-150 mutated *Oidiodendron maius*.**

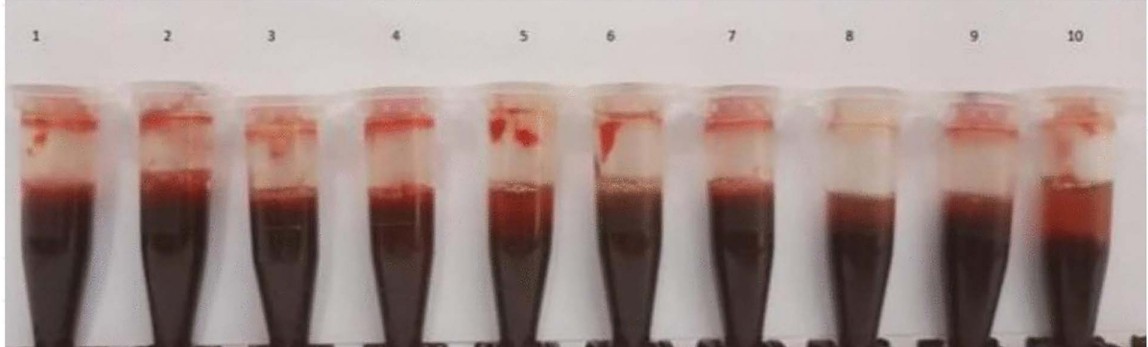

**Fig 5. Fibrinolytic activity of enzyme produced by BBTI-EB-150 mutant strain.**

## Kinetic and Thermodynamic analysis

A number of kinetic parameters such as pH, temperature, metal ions, substrate concentration, and Activation energy ($E_a$) were assessed to determine their influence on the activity of the fibrinolytic enzymes.

## Effect of pH on Enzyme Activity

The activity of purified fibrinolytic enzymes was across a diverse range of pH values (4.0–8.0) to identify the optimum pH for enzyme functionality. Both wild-type *O. maius* and its ethidium bromide mutant strain exhibited maximum enzyme activity at pH 7.5, recording activities of 305 and 425 U/mL, respectively. It was observed that enzyme activity increased progressively until pH 7.5 and declined beyond this point (Fig 7A, 7B).

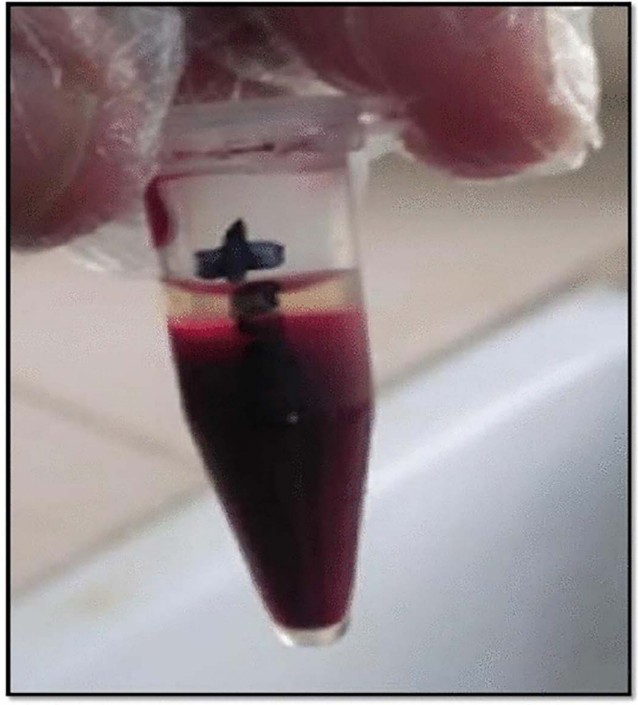

**Fig 6. CaCl$_2$-induced clotting time assay for ethidium bromide mutant strain (BBTI-EB-150) of *Oidiodendron maius*.**

**Table 10. CaCl$_2$ Induced clotting time assay for fibrinolytic enzymes from *Oidiodendron maius*.**

| Source of Fibrinolytic Enzymes | Anticoagulant activity |
|---|---|
| **Blank** | – |
| **Standard EDTA** | +++++++ |
| ***Oidiodendron maius* BBTI-Wild** | ++++ |
| ***Oidiodendron maius* BBTI-UV-50** | +++ |
| ***Oidiodendron maius* BBTI-G-120** | +++ |
| ***Oidiodendron maius* BBTI-EB-150** | ++ |
| ***Oidiodendron maius* BBTI-NA-60** | ++++ |

*No clotting: +++++++, Clotting after 20 min: +++, Clotting after 25 min: ++.

## Effect of Temperature on Enzyme Activity

Temperature is major factor influencing enzymes activity. The enzymes produced from wild-type and ethidium bromide-mutant strains of *O. maius* demonstrated maximum activity at 40°C and 35°C, respectively, with enzyme activities recorded as 305 U/mL and 425 U/mL (Fig 7C and 7D).

## Effect of metal ions on enzyme activity

The effects of various metal ion on the activity of fibrinolytic enzymes were investigated. For the enzyme produced by ethidium bromide mutant strains of *O. maius*, it was observed that $Cu^{2+}$ and $Zn^{2+}$ enhances enzyme activity, whereas $Mg^{2+}$ inhibited it.

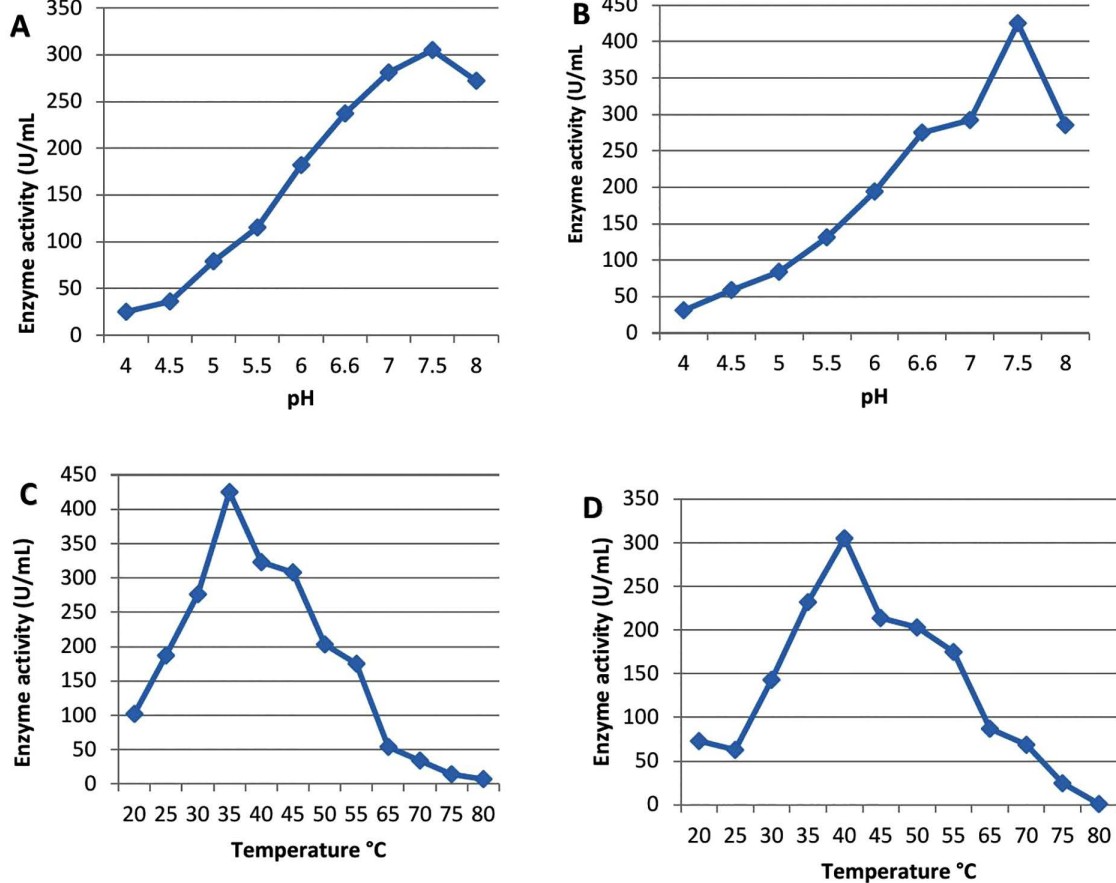

**Fig 7. A)** Effect of pH on enzyme activity from wild-type strain of *Oidiodendron maius* **B)** Effect of pH on enzyme activity from Ethidium bromide strain of *Oidiodendron maius* **C)** Effect of temperature on enzyme activity from wild strain of *Oidiodendron maius* **D)** Effect of temperature on enzyme activity from Ethidium bromide mutated strain of *Oidiodendron maius*.

## Determination of Enzyme Inhibitors for the Inhibition of Fibrinolytic Enzyme Production

The effect of various enzyme inhibitors (PMSF, 1–10, Phenanthroline, p-Chloromercuryl benzoate, EDTA, detergent (SDS (1%)), bleaching agent ($H_2O_2$ (1%)) and salt solution (NaCl)) were studied on the fibrinolytic enzymes. It was observed that EDTA and 1, 10- phenanthroline significantly inhibited the activity of the enzyme derived from the ethidium bromide mutant strain of *O. maius*.

## Determination of Km and Vmax

Using the Lineweaver-Burk method, the effects of varying substrate concentrations on enzyme activity were assayed, and the Michaelis-Menten kinetic constants were calculated. Km and Vmax values were calculated for both wild-type and the ethidium bromide-derived mutant BBTI-EB-150 of *O. maius*. The mutant-derived enzyme showed a Km of 28.13±2.0 µM and a *Vmax* of 42±5.2, with a corresponding specificity constant of 1718.93 s$^{-1}$ mM$^{-1}$.

## Thermodynamics of Irreversible Thermal Inactivation

To evaluate thermostability, a known concentration of enzyme was incubated at temperature ranging from 30 to 90°C in the absence of substrate. The enthalpy of denaturation (ΔH*) for the fibrinolytic enzyme from the ethidium bromide mutant

of *O. maius* was recorded 35.12 kJ/mol at 40ºC with an increasing trend observed as temperature rose. The free energy of denaturation (ΔG*) was found to be 91.23 kJ/mol at 40ºC. Negative entropy (ΔS*) values at various temperatures suggested the thermodynamic stability of the enzymes produced from mutant strain (Table 11).

## Discussion

The primary aim of this project was to induce mutations in a local strain of *O. maius* using both chemical and physical mutagens to enhance the production and activity of fibrinolytic enzymes. Wheat bran, as a substrate proved to be economically viable and effective for supporting fungal growth and enzyme production. This study reinforces the potential of *O. maius*, especially mutated strains, for the hyperproduction of fibrinolytic enzymes.

Two isolates, designated Bahawalpur Bio-Thrombolytic Isolate 1 and 4 (BBTI-1 and BBTI-4) demonstrated unique and promising characteristics. When grown on fibrin basal salt medium supplemented with wheat bran, exhibited robust growth and high fibrinolytic enzyme activity. Notably, wheat bran served as an efficient and cost-effective substrate, eliminating the need for expensive inputs. These results highlight the industrial-scale applicability of these isolates, particularly in resource-limited settings.

Kill curve analysis revealed that different mutagens required varied optimal exposure conditions to achieve effective mutation without complete loss of viability. The selected doses, 120 kRad gamma irradiation, 50 minutes of UV exposure, 150 minutes of ethidium bromide treatment, and 60 minutes of nitrous acid application, resulted in effective 3-log kills. These parameters were ideal for selecting enhanced mutants and align with the findings of Sobrun *et al.* (2012) [25], supporting the reliability of the mutagenesis protocols.Critical parameters such as pH, temperature, inoculum size and substrate concentration significantly influenced fermentation. Our wild and mutated strains gave maximum enzyme activity at pH 7.0 to 8.0. Shilpa H.K. (2017) emphasized the importance of maintaining optimal pH at to maximize enzyme yields [14]. Since, microbial growth and enzyme synthesis are highly pH dependent, even slight deviations from the optimal pH significantly reduced enzyme production. These results agree with Madhuri *et al.*, (2011) [26], Patel *et al.,* (2011) [27], El-Mongy and Taha (2012) [15], Jasim and Ali (2021) [28],Faran *et al.*, (2015) [29], Deng *et al.*, (2018) [30], Cardoso *et al.*, (2022) [31] and Kunhiraman (2024) [32] all of whom reported optimal pH values ranging between 7.0 to 8.0. Maximum enzyme yield of 336.91U/mL was observed from ethidium bromide at 37°C and significant decline in enzyme production was recorded when deviated from this temperature. This finding is in strong correlation with the results reported by Nascimento *et al.,* (2016) who also reported 37°C as the optimum temperature for fibrinolytic enzyme produced by *Mucor subtilissimus* [33]. Dubey *et al.,* (2011) also demonstrated optimal streptokinase yield at 37°C via media optimization, which aligns with our findings.. Additionally, El-Mongy and Taha (2012) reported an enzyme yield of 91.6 U/mL at 37°C, further validating the results of the current study [28]. Results reported by Cardoso *et al.,* (2022) [31] also strengthen our study.

**Table 11. Kinetic and thermodynamic parameters for irreversible thermal inactivation of fibrinolytic enzyme derived from BBTI-EB of *Oidiodendron maius*.**

| Temperature (K) | Kd (min) | t1/2 (min) | ΔH* (KJ /mol) | ΔG* (KJ /mol) | ΔS* (J mol/k) |
|---|---|---|---|---|---|
| 313 | 3.9 | 131 | 36.22 | 96.49 | −186.32 |
| 318 | 3.5 | 204 | 38.41 | 99.56 | −196.31 |
| 323 | 4.9 | 137 | 39.37 | 100.41 | −192.41 |
| 328 | 5.18 | 149 | 40.54 | 101.24 | −192.93 |
| 333 | 8.92 | 105 | 41.39 | 97.92 | −187.33 |
| 338 | 9.69 | 89 | 41.43 | 100.74 | −192.03 |
| 343 | 12.1 | 58 | 42.38 | 101.42 | −194.44 |
| 348 | 13.4 | 51 | 42.63 | 102.94 | −195.12 |
| 353 | 21 | 26 | 44.62 | 103.02 | −195.94 |

Enzyme synthesis by wild-type and the mutant strains of *O. maius* were observed to increase with the addition of 4.5% inoculum size. Similarly, Anusree *et al*., (2020) observed that increasing the inoculum size positively influenced the rate of enzyme production. According to our study, fermentation duration of 36 hours was found to be the most effective for enzyme synthesis in the majority of mutant strains. This is notably shorter than the 75-hour fermentation period reported by Dubey *et al.*, (2011) for maximum urokinase synthesis [23]. Similarly, Madhuri *et al*., (2011) reported that a 24-hour fermentation period was sufficient to produce maximum streptokinase [26]. El-Mongy and Taha (2012), however, recorded 72 hours as the optimum period for producing 91.6 U/mL of streptokinase [15]. These comparisons indicate that the optimized fermentation durations in the current study are relatively shorter and more efficient.

The results indicate effective removal of salts and impurities via dialysis, enhancing both enzyme activity and specific activity. The highest purification fold and recovery were observed in the ethidium bromide treated strain. These findings are in line with those reported by Rym Agrebi (2009), who purified enzymes from *Bacillus subtilis* A26 using ammonium sulfate precipitation followed by dialysis, achieving a 4–5 fold increase in purity [34]. Additionally, Seon *et al.,* (2011) demonstrated similar outcomes while purifying enzymes from *Cordyceps militaris* using ammonium sulfate precipitation [35]. These supporting studies affirm the efficiency of desalting as a step in enzyme purification.

The enhanced purification and activity levels achieved after ion exchange chromatography, particularly in the ethidium bromide mutant. Barros *et al.* (2020) reported a 32.42-fold purification and 7,988 U/mg specific activity for a fibrinolytic enzyme from *Arthrospira platensis* using 40–70% ammonium sulfate precipitation [36]. Similarly, Bajaj *et al.,* (2014) achieved a 4.8-fold purification and 10.4% recovery for a protease from *Bacillus subtilis* I-2 [37]. The gel filtration chromatography results demonstrate that mutant strains of *O. maius* significantly outperformed the wild-type strain in terms of enzyme activity, fold purification, and recovery percentage. Particularly, the BBTI-EB-150 strain produced the highest specific activity and purification fold. The findings are consistent with previous studies. Gopinath *et al.,* (2020) reported enhanced production of serrapeptase from *Serratia marcescens* using gel permeation chromatography with Sephadex, obtaining 12.3 mg/mL protein content, 17,877.31 U/mL enzyme activity, 1,453.44 U/mg specific activity, and 4.34-fold purification [38]. Babashamsi *et al.* (2009) reported up to 95% recovery using a combination of gel and ion exchange chromatography [39]. These previously reported findings strongly support the results of the current study.

The appearance of bands around 35 kDa suggests the presence of fibrinolytic enzymes. Urokinase exists in molecular form as 34.5 kDa. Based on this, the observed 35 kDa band is presumed to correspond to urokinase. This interpretation is supported by Tharwat (2006) [22]. Additionally, Yang *et al.* (2004) reported a 36 kDa proteolytic enzyme, synthesized by a different strain of *O. maius*, further corroborating the result [40].

The observed lytic activity confirms that the purified enzyme from BBTI-EB-150 strain possess significant fibrinolytic potential. The clot lysis percentages indicate effective hydrolysis of fibrin networks in the blood clots, suggesting that these mutant strains could serve as potential sources for thrombolytic agents. The findings suggest that pH 7.5 is the optimal value for maximum fibrinolytic enzyme activity from both wild and mutant strains. The enzyme activity declined beyond this optimum, consistent with the behavior of many fungal proteases. These results are in strong agreement with previously published findings of Gopinath *et al.,* (2020) [14] and Lusiana *et al.,* (2023) [41].

The results indicate that enzyme activity is strongly temperature-dependent, with peak fibrinolytic activity achieved within a moderate temperature range. These outcomes are consistent with those of Shilpa H.K. (2017), who studied proteolytic enzyme production from *Aspergillus* sp. and reported optimal enzyme activity at 40°C [36]. The results are also in line with the findings of Taneja *et al.,* (2017) who reported that the presence of certain metal ions increased enzyme stability and activity. The inhibition by EDTA and 1,10-Phenanthroline indicates that the enzyme is likely a metalloprotease, as both are known metal ion chelators. This observation is strongly supported by the findings of Zhao *et al*., (2022) and Wang *et al*. (2024), whose fibrinolytic enzymes produced by *Aspergillus versicolor* ZLH-1 and *Coprinus comatus* respectively, were also completely inhibited by EDTA.

 

Purified enzymes from mutant strains demonstrated better blood clot lysis potential, improved thermal stability, and greater overall efficiency, making them suitable candidates for clinical and industrial use. This study marks the first attempt to induce hyper production of fibrinolytic enzymes from *O. maius* through targeted mutagenesis. Among the four generated mutants, BBTI-EB-150 showed the highest promise due to its enhanced thermo-stability, lower nutrient requirements, and superior enzyme yield. The kinetic characterization of these enzymes, also performed for the first time in this context, adds novelty and reliability to the study. Overall, this research provides a foundational basis for developing cost effective production of thrombolytic enzymes from improved fungal strains for CVD treatment.

## Future prospects

The production of fibrinolytic enzymes from mutant strains of *O. maius* represents highly promising frontier in medicine. The future prospects of these enzymes lie in the advanced cardiovascular therapeutics, cost effective bio-factories and broad industrial applications. In future, further research can involve the *in vivo*experimentation to prove these enzymes to be effective in human blood clot lysis.

## Supporting information

**S1 File. SDS-PAGE analysis of fibrinolytic enzymes produced by BBTI-EB-150 mutated *Oidiodendron maius.*** (DOCX)

## Acknowledgments

The authors gratefully acknowledge the Institute of Biochemistry, Biotechnology and Bioinformatics at The Islamia University of Bahawalpur for providing the essential laboratory facilities and technical resources used to conduct this research.

## Author contributions

**Conceptualization:** Muhammad Iqbal, Gull-e-Faran.

**Formal analysis:** Hina Sajid, Gull-e-Faran.

**Investigation:** Hina Sajid, Muhammad Iqbal.

**Methodology:** Hina Sajid, Muhammad Iqbal, Gull-e-Faran.

**Resources:** Gull-e-Faran.

**Software:** Hina Sajid.

**Validation:** Hina Sajid.

**Visualization:** Hina Sajid.

**Writing – original draft:** Hina Sajid, Muhammad Iqbal, Gull-e-Faran.

**Writing – review & editing:** Hina Sajid, Muhammad Iqbal, Gull-e-Faran.

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
