## [Decision Letter · Decision Letter 0]

12 Dec 2025

Dear Dr. Gull-e-Faran,

Thank you for submitting your manuscript to PLOS ONE. After careful consideration, we feel that it has merit but does not fully meet PLOS ONE’s publication criteria as it currently stands. Therefore, we invite you to submit a revised version of the manuscript that addresses the points raised during the review process.

We look forward to receiving your revised manuscript.

Kind regards,

Faiz Ahmad Joyia, Ph.D.

Academic Editor

PLOS One

Journal Requirements:

4. In your Methods section, please provide additional information regarding the permits you obtained for the work. Please ensure you have included the full name of the authority that approved the field site access and, if no permits were required, a brief statement explaining why.

5. We note that your Data Availability Statement is currently as follows: All relevant data are within the manuscript and its Supporting Information files

6. Please ensure that you refer to Figure 1 in your text as, if accepted, production will need this reference to link the reader to the figure.

Reviewers' comments:

Reviewer's Responses to Questions

**Comments to the Author**

1. Is the manuscript technically sound, and do the data support the conclusions?

Reviewer #1: Yes

Reviewer #2: Yes

2. Has the statistical analysis been performed appropriately and rigorously?

Reviewer #1: Yes

Reviewer #2: Yes

3. Have the authors made all data underlying the findings in their manuscript fully available?

Reviewer #1: Yes

Reviewer #2: Yes

4. Is the manuscript presented in an intelligible fashion and written in standard English?

Reviewer #1: Yes

Reviewer #2: Yes

Reviewer #1: Enhanced Fibrinolytic Enzyme Production by Oidiodendron maius through Green Bioprocessing of Agro-Industrial residue

Reviewer’s Comments:

The manuscript presents an interesting study on improving fibrinolytic enzyme production using mutagenized strains of Oidiodendron maius, combined with low-cost agro-industrial substrates. The topic is relevant to biotechnology, enzyme engineering, and the development of eco-friendly bioprocesses for thrombolytic agents. The work includes extensive experimental data covering isolation, mutagenesis, fermentation optimization, purification, biochemical characterization, and kinetic/thermodynamic analysis.

While the study has clear novelty as particularly the use of O. maius mutants for enhanced fibrinolytic enzyme production. However, it requires improvement in scientific clarity, methodological detail, statistical rigor, language quality, and structural organization. The manuscript is far too lengthy and repetitive, and several data presentations need careful revision.

Overall, the manuscript shows potential but requires revision before publication.

1. Abstract section is too long and should be condensed.

2. Please include more recent citations in Introduction section as several citations are outdated.

3. The section on thrombotic diseases is disproportionately long relative to the focus of the study.

4. Provide details of instrumentation in materials and method section and also clarify buffer compositions.

5. Please report all biochemical tests briefly which were used for the confirmation of Oidiodendron maius.

6. What is the purpose of using wheat-bran? Why it has been used specifically? Elaborate its importance.

7. Why did the authors test the fungal capability to grow at 60 oC? Is it a thermophilic fungus?

8. What is the purpose of 40% and 60% sediments mentioned but not explained in the purification section?

9. Avoid repeating numerical values extensively in text and only mention in the tables or supplementary data.

10. In Discussion section, excessive citation of unrelated bacterial studies were observed, focus more on fungal systems.

11. The manuscript is excessively long and contains large sections of repeated explanations that could be condensed. The Discussion section repeats many Results with minimal interpretation.

12. Some references are duplicated.

13. What limitations exist in current fibrinolytic enzyme sources.

14. The mutagenesis procedures require clearer explanation of exposure conditions (UV wavelength inconsistently stated as 240 nm; standard UV mutagenesis uses 254 nm).

15. Authors should explicitly state the number of replicates per experiment.

16. All the figures should have proper labels, axis titles, units.

17. The purification tables contain inconsistencies between U/mL and U/mg.

18. The paper mentions multiple mutants were tested, but only the ethidium bromide mutant's data are highlighted in the abstract. A table of comparison of all the mutants, highlighting the best data of ethidium bromide mutant would strengthen the argument.

19. The manuscript contains numerous grammatical errors and unclear sentences.

Reviewer #2: The manuscript presents an extensive investigation into the enhancement of fibrinolytic enzyme production in Oidiodendron maius through mutagenesis, followed by purification and characterization of enzyme activity. The work is scientifically relevant, methodologically rich, and demonstrates a clear link between experimental findings and potential application in thrombolytic therapy. The novelty — particularly the first reported hyperproduction and kinetic profiling from mutated O. maius strains — is of significance.

However, the manuscript requires some editorial and structural refinement to improve clarity, cohesion, and scientific readability. The volume of information is high, with several sections appearing overly descriptive and repetitive. With revision, this work could become impactful within enzyme biotechnology and biomedical application domains.

The results contain large blocks of information without sectional breaks. This makes it difficult for readers to locate important findings. I suggest authors to make subsections such as:

• Mutagenesis optimization

• Fermentation parameter influence

• Purification profile of wild vs mutants

• Kinetic & thermodynamic analysis

• Clot lysis and biomedical relevance

There is strong literature support, but results and citations are sometimes interwoven in a way that disrupts flow. Recommended structure:

Result → Interpretation → Supporting literature, rather than alternating statements.

Language, clarity, and grammar

Several sentences are long, repetitive, or grammatically weak. Conciseness is needed. For example:

“Our results are revealed…” → “Our results revealed…”

“These findings correlate well…” appears repeatedly and could be merged.

A focused language edit would substantially improve readability.

The manuscript mentions novelty, but it would be strengthened by explicitly stating:

• How much improvement was achieved versus wild type

• What makes BBTI-EB superior mechanistically

• Next step towards scale-up / drug development

**Do you want your identity to be public for this peer review?** For information about this choice, including consent withdrawal, please see our Privacy Policy

Reviewer #1: No

Reviewer #2: **Yes:** Urmil Bansal

---

## [Author Response · Author response to Decision Letter 1]

28 Jan 2026

Enhanced Fibrinolytic Enzyme Production by Oidiodendron maius through Green Bioprocessing of Agro-Industrial residue

Authors are thankful to the reviewers and Editor for their comments to improve the quality of the manuscript. Authors reviewed the manuscript critically and tried their best to address and solve all the comments.

Reviewer #1:

Reviewer’s Comments:

Reviewer 1:

The manuscript presents an interesting study on improving fibrinolytic enzyme production using mutagenized strains of Oidiodendron maius, combined with low-cost agro-industrial substrates. The topic is relevant to biotechnology, enzyme engineering, and the development of eco-friendly bioprocesses for thrombolytic agents. The work includes extensive experimental data covering isolation, mutagenesis, fermentation optimization, purification, biochemical characterization, and kinetic/thermodynamic analysis.

While the study has clear novelty as particularly the use of O. maius mutants for enhanced fibrinolytic enzyme production. However, it requires improvement in scientific clarity, methodological detail, statistical rigor, language quality, and structural organization. The manuscript is far too lengthy and repetitive, and several data presentations need careful revision.

Overall, the manuscript shows potential but requires revision before publication.

Comment 1: Abstract section is too long and should be condensed.

Response: Abstract has been reviewed and condensed as per Reviewer’s suggestion.

Comment 2: Please include more recent citations in Introduction section as several citations are outdated.

Response: Outdated citations have been eliminated from the introduction section and the recent references have been added.

Comment 3: The section on thrombotic diseases is disproportionately long relative to the focus of the study.

Response: To maintain the focus on fibrinolytic enzyme production from the fungal strain, the paragraph giving more details about irrelevant studies has been minimized.

Comment 4: Provide details of instrumentation in materials and method section and also clarify buffer compositions.

Response: Instrumentation and chemical details have been described in the text wherever necessary and buffer composition has also been mentioned in the materials and methods section.

Comment 5: Please report all biochemical tests briefly which were used for the confirmation of Oidiodendron maius.

Response: Appropriate biochemical tests employed for fungal confirmation are added as per required at appropriate positions in the manuscript.

Comment 6: What is the purpose of using wheat-bran? Why it has been used specifically? Elaborate its importance.

Response: Wheat-Bran is considered as priority substrate for fungal cultivation due to the reasons:

1. It is a complete package which provides every nutrient a fungus needs to thrive.

2. It has an ideal Carbon and Nitrogen ratio as compared to other alternatives like wheat- straw or sawdust.

3. It is also rich in essential minerals and vitamins.

4. Moreover, it contains residual starch which is easily accessible source of energy for fungus. Furthermore, it has complex composition containing hemicellulose, cellulose and proteins that tricks the fungus into turning on a wide array of genes to produce a variety of enzymes.

5. It is produced in massive quantities as a by-product, which makes it readily available and cheaper substrate. Moreover, the relevant justification is also added in the manuscript at appropriate positions.

Comment 7: Why did the authors test the fungal capability to grow at 60o C? Is it a thermophilic fungus?

Response: Some of the species of the genus Oidiodendron e.g. Oidiodendron flavum are thermophilic and they can live as well as produce enzymes at such elevated temperatures. So, there was a probability of the fungus and its four mutants to grow and produce enzymes at 60o C. Hence, the authors designed their parameter optimization protocol to check this probability.

Comment 8: What is the purpose of 40% and 60% sediments mentioned but not explained in the purification section?

Response: In purification step, 40% and 60% saturation levels of ammonium sulfate are used for the purpose of fractional precipitation. It allows the protein separation based on their differing solubility’s. In the first step, ammonium sulfate is added to reach its 40% saturation that allows the removal of large or less soluble proteins along with some nucleic acids. They settle down as precipitates and the target protein may still remain in the supernatant as it is still soluble at 40%. Then more ammonium sulfate is added to the 40% supernatant to bring the total concentration to 60% saturation which precipitates the target protein in the pellet.

Comment 9: Avoid repeating numerical values extensively in text and only mention in the tables or supplementary data.

Response: Repetition of the numerical values in the text has been minimized as suggested by the reviewer.

Comment 10: In Discussion section, excessive citation of unrelated bacterial studies was observed, focus more on fungal systems.

Response: Some recent citations referring to the fungal studies has been added but a few findings related to the fibrinolytic enzyme produced by bacteria are still cited as they support studies related to the thermodynamic properties of enzyme itself.

Comment 11: The manuscript is excessively long and contains large sections of repeated explanations that could be condensed. The Discussion section repeats many Results with minimal interpretation.

Response: The discussion section has been condensed and improved scientifically explaining the findings, previous references and the interpretation. Moreover, the manuscript is trimmed where required as more trimming will damage the soul of the manuscript.

Comment 12: Some references are duplicated.

Response: This issue has been resolved by the authors by deleting the duplicated references after cross checking.

Comment 13: What limitations exist in current fibrinolytic enzyme sources?

Response: There are some production challenges which include the stability issues i.e. sensitivity towards pH and temperature.

Comment 14: The mutagenesis procedures require clearer explanation of exposure conditions (UV wavelength inconsistently stated as 240 nm; standard UV mutagenesis uses 254 nm).

Response: For physical mutagenesis, UV irradiation was performed at 240 nm and 254 nm for selecting the optimum wavelength. It was observed by authors that exposure of fungal strains at 240 nm wavelength gave higher killing rates of fungal strains, whereas exposure at 254 nm wavelength was highly efficient in terms of mutagenesis. So, the wavelength of 254 nm was used for UV mutagenesis of Oidiodendron maius.

Comment 15: Authors should explicitly state the number of replicates per experiment.

Response: All the experiments were performed in triplicates independently and are mentioned in the text.

Comments 16: All the figures should have proper labels, axis titles, units.

Response: Figures are rechecked for proper labels, axis titles and units as per suggested by the reviewer.

Comment 17: The purification table contains inconsistencies between U/mL and U/mg.

Response: Inconsistency between units in purification table has been resolved by the authors.

Comment 18: The paper mentions multiple mutants were tested, but only the ethidium bromide mutant's data are highlighted in the abstract. A table of comparison of all the mutants, highlighting the best data of ethidium bromide mutant would strengthen the argument.

Response: The ethidium bromide mutated strain showed good results in the current study as compared to the wild and mutated strains. Therefore, author mentioned them in abstract. The differences between the results of ethidium bromide mutated and the rest of the strains have been addressed in figure 2, 3, 7 and table 3, 9, 10 in the manuscript.

Comment 19: The manuscript contains numerous grammatical errors and unclear sentences.

Response: The manuscript has been revised focusing extensively on resolving grammatical errors and verbal mistakes in sentences.

Reviewer 2:

The manuscript presents an extensive investigation into the enhancement of fibrinolytic enzyme production in Oidiodendron maius through mutagenesis, followed by purification and characterization of enzyme activity. The work is scientifically relevant, methodologically rich, and demonstrates a clear link between experimental findings and potential application in thrombolytic therapy. The novelty — particularly the first reported hyper-production and kinetic profiling from mutated O. maius strains — is of significance.

However, the manuscript requires some editorial and structural refinement to improve clarity, cohesion, and scientific readability. The volume of information is high, with several sections appearing overly descriptive and repetitive. With revision, this work could become impactful within enzyme biotechnology and biomedical application domains.

Comment 1: The results contain large blocks of information without sectional breaks. This makes it difficult for readers to locate important findings. I suggest authors to make subsections such as:

• Mutagenesis optimization

• Fermentation parameter influence

• Purification profile of wild vs mutants• Kinetic & thermodynamic analysis

• Clot lysis and biomedical relevance

Response: Authors have resolved the issue by adding sectional breaks in results and also the suggested subsections are added in the results section as per the valuable suggestions by reviewer.

Comment 2: There is strong literature support, but results and citations are sometimes interwoven in a way that disrupts flow. Recommended structure:

Result → Interpretation → Supporting literature, rather than alternating statements.

Response: Authors have followed the recommended structure for the results, interpretation and supporting literature by the reviewer. Authors have worked on clarity of results presentation along with the supporting literature.

Comment 3: Several sentences are long, repetitive, or grammatically weak. Conciseness is needed. For example:

“Our results are revealed…” → “Our results revealed…”

“These findings correlate well…” appears repeatedly and could be merged.

A focused language edit would substantially improve readability.

Response: The manuscript has been reedited and improved to overcome repetition and grammatical errors as suggested by the reviewers.

Comment 4: The manuscript mentions novelty, but it would be strengthened by explicitly stating:

1. How much improvement was achieved versus wild type

Response: Improvements in specific activities, enzyme yields, fold purification, clot lysis ability and the other improvements in the mutants as compared to the wild O. maius have been mentioned under respective headings in results section explicitly.

2. What makes BBTI-EB superior mechanistically

Response: When compared with the other mutated strains, BBTI-EB proved itself to be superior clearly by giving higher enzyme yield. It provided several mechanistic advantages in the laboratory. Mutagenesis of O. maius with ethidium bromide enhances the extracellular enzyme secretion efficiency of fungus and its capability of exporting large protease molecules out of the cell and surrounding medium increase. This makes the downstream processing of enzyme much easier and cheaper. Mechanistically, this strain can produce the enzyme at a higher flux as compared to the wild or other mutated strains. The enzymes produced show higher substrate specificity for fibrin and they often act as plasmin-like proteases.

3. Next step towards scale-up / drug development.

Response: Future prospects are added under the discussion section as per suggestion.

Dear Editor,

The SDS-PAGE analysis was conducted in 2023 following standardized protocols. Due to a critical hardware failure of the gel imaging system during data acquisition, the raw digital files were unrecoverable. Consequently, the results are presented via the two high-resolution photographs, one in the main manuscript body and the second one as supporting information, captured prior to the system malfunction. The integrity of the samples and the migration patterns remain consistent with expected experimental outcomes.

---

## [Decision Letter · Decision Letter 1]

2 Mar 2026

Enhanced Fibrinolytic Enzyme Production by Oidiodendron maius through Green Bioprocessing of Agro-Industrial residue

PONE-D-25-56032R1

Dear Dr. Gulle-e-Faran,

We’re pleased to inform you that your manuscript has been judged scientifically suitable for publication and will be formally accepted for publication once it meets all outstanding technical requirements.

Kind regards,

Faiz Ahmad Joyia, Ph.D.

Academic Editor

PLOS One

Additional Editor Comments (optional):

Reviewers' comments:

Reviewer's Responses to Questions

**Comments to the Author**

Reviewer #3: (No Response)

Reviewer #4: (No Response)

2. Is the manuscript technically sound, and do the data support the conclusions?

Reviewer #3: Yes

Reviewer #4: Yes

3. Has the statistical analysis been performed appropriately and rigorously?

Reviewer #3: Yes

Reviewer #4: Yes

4. Have the authors made all data underlying the findings in their manuscript fully available?

Reviewer #3: Yes

Reviewer #4: Yes

5. Is the manuscript presented in an intelligible fashion and written in standard English?

Reviewer #3: Yes

Reviewer #4: Yes

Reviewer #3: The manuscript presents extensive work with promising enhancement of fibrinolytic enzyme production. Purification results presented in the manuscript are potentially impactful for enzyme biotechnology. However, to make it readable and reliable for publication it requires some changes

Comment 1: In Methods it is mentioned that production flasks were incubated at 37°C/120 rpm for 24 days, but in same section and Results, fermentation time is written as 12–72 h with maxima at 24–36 h, creating a major inconsistency. Please confirm the actual fermentation duration used for enzyme production and revise the manuscript accordingly.

Comment 2: UV mutagenesis parameters are inconsistent and include an implausible wavelength “25440 nm” and exposure time is reported differently (e.g., 1 hour vs 50 minutes). There is confusion in text, either 240 or 254 nm. Please provide one definitive UV protocol (wavelength, lamp type, intensity/dose, distance, exposure time/intervals, shielding) and keep it consistent across sections of article.

Comment 3: The statistical analysis section seems too vague, its two-factor randomized design with ANOVA, but does not specify the factors tested, whether interactions were evaluated, what post-hoc multiple-comparison procedure was used, how ANOVA assumptions were checked, or whether “triplicates” are biological vs technical replicates. Please revise to report n for each experiment, the exact ANOVA model (strain × parameter level), interaction results, post-hoc method (e.g., Tukey), assumption checks, and make sure every figure clearly states mean ± SD/SE and what the error bars represent.

Comment 4: The study measures fibrinolysis using fibrin-plate/punch-hole clear zone diameters yet reports activity as U/mL without defining what 1 unit represents. Please standardize the activity reporting by explicitly defining 1 U (and the calculation for U/mL), including any standard curve/reference enzyme, or add a quantitative spectrophotometric/chromogenic assay to support diffusion-based measurements.

Comment 5: The study is framed as green bioprocessing, but the strain improvement relies on ethidium bromide mutagenesis. Its better to either restrict “green” to the agro-industrial residue substrate aspect or tone down the green claim and clearly acknowledge this limitation. Please also add a brief biosafety and waste-disposal compliance statement describing how ethidium bromide was handled, decontaminated and disposed of according to institutional regulations.

Comment 6: Please ensure all figures have complete axis titles and units.

Comment 7: It is mentioned that bands from SDS-PAGE were excised. After removal, these bands were used for downstream applications.

Comment 8: Ensure consistency in mutant naming (BBTI-EB-150 / BBTI-E / etc.) across text, tables and figures.

Reviewer #4: The manuscript presents an interesting and timely study focusing on sustainable enzyme production using agro-industrial waste substrates. The topic is scientifically relevant, particularly in the context of green biotechnology, value-added bioprocessing, and the growing demand for fibrinolytic enzymes with potential therapeutic applications. The approach of utilizing agro-industrial residues aligns well with current trends in sustainable biomanufacturing and circular bioeconomy principles.

Overall, the study addresses a meaningful research question and has the potential to contribute to the field of industrial microbiology and enzyme biotechnology. However, several aspects of the manuscript require clarification, methodological refinement, and language improvement to enhance its scientific rigor and presentation quality.

1. Please include at least two additional relevant keywords in the Abstract to improve indexing and search visibility.

2. The last two paragraphs of the Introduction contain several grammatical issues. Careful language editing is required to improve clarity and readability.

3. In the Materials and Methods section, kindly recheck the names of the chemicals and their respective compositions for accuracy and consistency.

4. The rationale for selecting wheat bran as the preferred agro-industrial residue should be clearly justified. Please elaborate on its selection criteria and relevance to this study.

5. Please specify the type and molecular weight cut-off of the dialysis membrane used for desalting.

6. The subsection titled “Ammonium sulfate precipitation for partial purification” in the Results section requires revision to minimize grammatical and scientific inaccuracies. The description should be made more precise and technically sound.

7. The Discussion section would benefit from improvement in scientific language and deeper critical interpretation of the results in relation to existing literature.

**Do you want your identity to be public for this peer review?** For information about this choice, including consent withdrawal, please see our Privacy Policy

Reviewer #3: No

Reviewer #4: **Yes:** Sher Muhammad

---

## [Editor Report · Acceptance letter]

PONE-D-25-56032R1

PLOS One

Dear Dr. .,

I'm pleased to inform you that your manuscript has been deemed suitable for publication in PLOS One. Congratulations! Your manuscript is now being handed over to our production team.

Kind regards,

on behalf of

Dr. Faiz Ahmad Joyia

Academic Editor

PLOS One